# ChatVLA-2:
# Vision-Language-Action Model with Open-World Embodied Reasoning from Pretrained Knowledge

**Zhongyi Zhou**[1][*]  **Yichen Zhu**[2][*][†]  **Xiaoyu Liu**[2]  **Zhibin Tang**[2]  **Junjie Wen**[2]
**Yaxin Peng**[3]  **Chaomin Shen**[1][†]  **Yi Xu**[2]

[1] East China Normal University  [2] Midea Group  [3] Shanghai University

**chatvla-2.github.io**

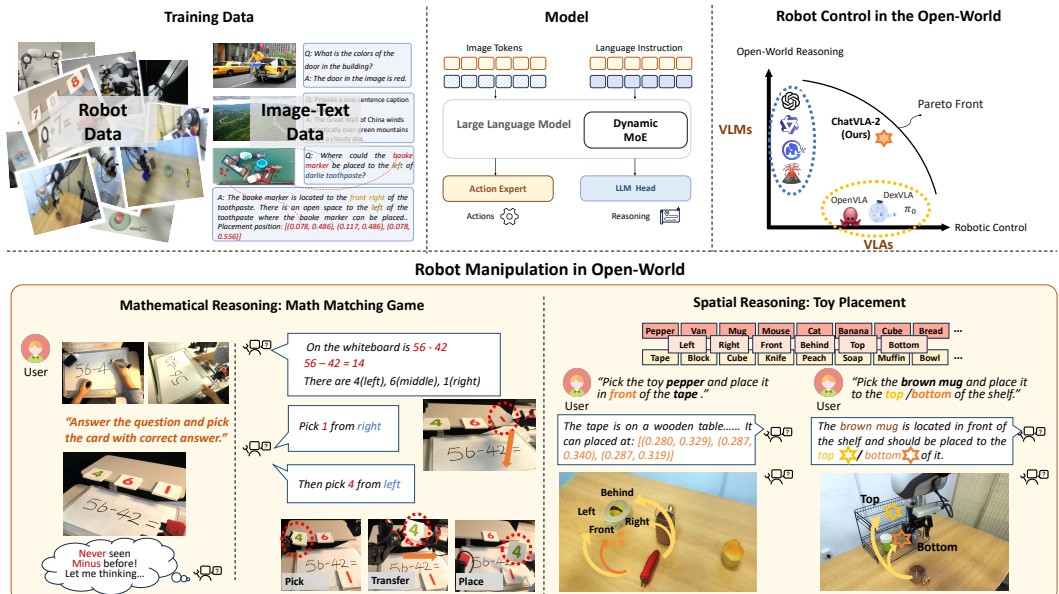

Figure 1: **Our proposed ChatVLA-2 model enables generalized open-world reasoning and reasoning following abilities.** We designed two tasks—a math matching game and a toy placement experiment—to demonstrate its generalization ability.

## Abstract

Vision-language-action (VLA) models have emerged as the next generation of models in robotics. However, despite leveraging powerful pre-trained Vision-Language Models (VLMs), existing end-to-end VLA systems often lose key capabilities during fine-tuning as the model adapts to specific robotic tasks. We argue that a generalizable VLA model should retain and expand upon the VLM's core competencies: 1) **Open-world embodied reasoning** - the VLA should inherit the knowledge from VLM, i.e., recognize anything that the VLM can recognize, be capable of solving math problems, and possess visual-spatial intelligence, 2) **Reasoning following** – effectively translating the open-world reasoning into actionable

---

[*]Equal Contribution. Work done during Zhongyi Zhou's internship at Midea Group.
[†]Corresponding Authors.

steps for the robot. In this work, we introduce **ChatVLA-2**, a novel mixture-of-expert VLA model coupled with a specialized two-stage training pipeline designed to preserve the VLM's original strengths while enabling actionable reasoning. To validate our approach, we design a math-matching task wherein a robot interprets math problems written on a whiteboard and picks corresponding number cards from a table to solve equations. Remarkably, our method exhibits exceptional mathematical reasoning and OCR capabilities, despite these abilities not being explicitly trained within the VLA. Furthermore, we demonstrate that the VLA possesses strong spatial reasoning skills, enabling it to interpret novel directional instructions involving previously unseen objects. Overall, our method showcases reasoning and comprehension abilities that significantly surpass state-of-the-art imitation learning methods such as OpenVLA, DexVLA, and $\pi_0$. This work represents a substantial advancement toward developing truly generalizable robotic foundation models endowed with robust reasoning capacities.

# 1 Introduction

*If I have seen further, it is by standing on the shoulders of giants.*

Isaac Newton

Vision-language-action models (VLAs) have become a popular approach for tasks in robotics manipulation, navigation, and even full-body control. They have demonstrated remarkable capabilities in learning dexterous manipulation, tackling long-horizon tasks [1, 2], and enabling open-world generation [3, 4]. The success of VLAs, in contrast to traditional imitation learning methods, lies in their integration of pre-trained Vision-Language Models (VLMs). By leveraging the mature neural architectures from language models and multimodal networks, along with advanced training techniques and pre-trained knowledge from VLMs, VLAs significantly enhance robotic learning. This allows robots to better understand and interact with the world while improving their ability to perform complex physical tasks.

Intuitively, pre-training a VLA model consists of a powerful, pre-trained VLMs, such as PaliGemma [5] or Qwen-VL [6], should equip the robot with not only stronger vision-language feature embeddings but also the comprehensive capabilities inherent to VLMs — including recognizing everyday objects, reasoning about spatial relationships, and solving mathematical problems. Consider a simple task: writing down the answer to the equation $10 + 11 =$. Such a task is trivially easy for humans. A conventional hierarchical model would first leverage a pre-trained VLM to produce the answer (21), then invoke a low-level policy network to physically write it down. However, why might a VLA model struggle with such a simple task if it has never encountered the specific equation in its training data? In practice, fine-tuning on robotics-specific datasets often leads to the erosion of the original pre-trained knowledge from the VLM. For example, ChatVLA [7] illustrates that adapting a VLA model specifically for robotic control can cause previously acquired general knowledge to degrade significantly. As a result, the VLA model may fail to accomplish tasks that seem trivial to humans, simply because these tasks were absent from the training dataset.

Such a gap leads to a natural question: *How can we build VLA models that both keep their VLM prior intact and actively leverage it to achieve superior generalization in robotic control?*

In this study, we introduce ChatVLA-2, a significant advancement toward achieving a truly generalizable robotic foundation model. The goal of ChatVLA-2 is not to construct an omnipotent robot model capable of executing every conceivable task. Instead, our primary objective is to demonstrate the feasibility of leveraging the pre-trained knowledge embedded within the VLM backbone. By doing so, we enable end-to-end robotic systems to generalize across diverse tasks, which traditionally require explicit planning by an external agent. We argue that this generalization can be achieved by adhering to two fundamental principles:

- **Identifying overlapping feature spaces between multimodal understanding and robot control.** Image-text data and robotic control data generally reside in distinct feature spaces, often resulting in competition for shared parameter spaces within models. ChatVLA addresses this by employing separate static experts—one dedicated to multimodal understanding and another specialized for robotic control—to ensure the clear separation of these tasks

into distinct feature spaces. This separation allows VLA models to excel independently in both domains. However, the isolated nature of these feature spaces currently limits the transfer of pre-trained knowledge to robotic control tasks. If mutual beneficial features could be effectively preserved and distinct task-specific features disentangled, the VLA model would be better positioned to intuitively leverage its pre-trained knowledge, thus significantly enhancing its generalization capability in robotic control.

- **Ensuring VLA models act according to their internal reasoning.** Although VLA models demonstrate the capability for sophisticated internal reasoning, it remains uncertain whether their generated robotic actions accurately reflect this internal thought process. Previous research indicates that even large language models frequently produce outputs inconsistent with their thinking process [8, 9]. By ensuring that the action outputs through VLA models reliably follow their reasoning processes, we can substantially enhance their ability to generalize effectively across diverse and previously unseen tasks.

To achieve this, we propose a novel VLA model architecture employing a dynamic mixture-of-experts within the VLM backbone. This design explicitly disentangles the feature spaces related to multimodal understanding and robotic action while adaptively identifying and preserving their shared representations. Additionally, we introduce a straightforward reasoning-enhancement module designed to align the action expert's output more closely with the model's internal reasoning process. Furthermore, we implement a two-stage training strategy: The initial stage aims to preserve pre-trained multimodal knowledge, simultaneously training robotic actions and establishing connections between these components. During the second stage, the VLM backbone is frozen, and only the action expert remains trainable, explicitly enabling it to learn to generate actions consistent with the internal reasoning derived from the upper levels of the model.

To demonstrate the open-world reasoning and understanding capabilities of ChatVLA-2, we designed two tasks: a math matching game and a toy placement experiment. In the math matching game, we placed a whiteboard in front of the robot and wrote down a mathematical equation for the robot to solve. Several potential answers were placed before the robot, from which it had to select the correct solution and place it on the whiteboard. Importantly, we evaluated the robot entirely on out-of-distribution scenarios, meaning the presented equations never appeared in the training dataset. For evaluating spatial reasoning, we conducted a toy placement experiment. In this task, the robot was instructed to pick up a toy and place it at specific positions relative to various reference objects (e.g., to the right, left, front, behind, top, or bottom of objects). Many of the objects and directional instructions were entirely unseen during training. Therefore, this task required the model to accurately interpret the visual scene, reason about novel spatial instructions, and execute appropriate actions. Our experiments clearly illustrate the superior generalization capabilities of ChatVLA-2, particularly in reasoning and understanding tasks, surpassing existing imitation-learning approaches such as OpenVLA [10], DexVLA [2], and $\pi_0$ [1]. This work represents a significant step toward the development of truly generalizable robotic foundation models that transcend the limitations of fine-tuning data by effectively leveraging pre-trained VLM knowledge.

## 2 Related Work

**Vision-language-action models in robot learning.** Vision-language-action models (VLAs) form a growing body of research within imitation learning [11, 12, 13, 14, 15, 16, 17, 18, 19, 20, 21, 22, 23, 24] that leverages pre-trained vision-language models (VLMs) as a backbone to enable both language comprehension and observational understanding. These methods typically fine-tune large pre-trained VLMs to predict robot actions [25, 26, 27, 28, 29, 30, 31, 32, 33, 34, 35, 36, 37, 38, 39, 40, 41, 42, 43, 44, 45, 46, 47, 48, 49, 50, 51, 52, 53, 39, 54, 55]. These methods have demonstrated strong performance across various simulated and real-world tasks, covering diverse robotic embodiments such as bimanual robots, mobile manipulators, legged robots, and humanoids. They also exhibit generalization capabilities across different environments and various objects. However, existing VLA models still lack the ability to generalize beyond the scope of their training data. Despite incorporating pretrained vision-language models (VLMs) as their backbone, current VLA approaches fail to effectively utilize the pretrained knowledge from these VLMs, limiting robots' capabilities for open-world manipulation. Consequently, this significantly undermines the rationale behind employing pretrained VLMs within large-scale models. In this paper, we introduce ChatVLA-2, a novel model designed specifically to retain and leverage pretrained VLM knowledge, thus enabling

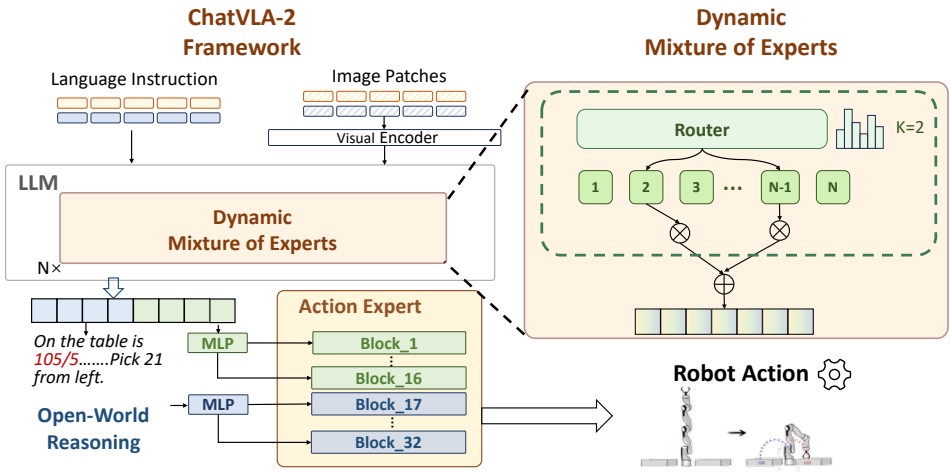

Figure 2: **Model architecture. Left**: A reasoning-following enhancement module is incorporated to ensure that the VLA model adheres to logical reasoning when performing actions. **Right**: Our method leverages a dynamic mixture-of-experts architecture to disentangle conflicting features between multimodal understanding and robotic control, while effectively integrating mutually beneficial features.

robots to perform open-world tasks effectively through pretrained reasoning and extensive general knowledge.

**Embodied Reasoning in VLA models.** A substantial amount of research has been dedicated to enhancing vision-language-action (VLA) models by incorporating the chain-of-thought (CoT) [56] methodology, inspired by the recent successes of large language models (LLMs) in various cognitive and reasoning tasks. The primary motivation behind adopting CoT is to replicate the sophisticated reasoning and decision-making capabilities of LLMs within robotic systems, enabling robots to perform more complex, context-aware actions in dynamic, real-world environments. For instance, Embodied-CoT [57] and CoA-VLA [58] utilize structured textual instructions enriched with spatial localization information, CoT-VLA [59]/VPP [60] integrates reasoning via generated visual imagery, DiffusionVLA [61], DexVLA [2], and $\pi_{0.5}$ [3] rely on plain language instructions. However, in these models, reasoning—whether represented through textual instructions or visual cues—is explicitly trained and consequently limited to knowledge contained within the training datasets, restricting their capacity for broader generalization. In this work, we significantly advance this line of research by leveraging pretrained knowledge from VLMs, thereby empowering VLA models with enhanced open-world reasoning and generalization capabilities.

## 3   Methodology

This section introduces our proposed ChatVLA-2 and is organized into three parts. Section 3.1 provides preliminary background on vision-language-action (VLA) models. Section 3.2 details the neural architecture, and Section 3.3 presents the two-stage training strategy. Together, these components empower the VLA model with open-world reasoning and understanding capabilities.

### 3.1   Preliminary: Vision-Language-Action Model

VLA models, leveraging pre-trained VLM perception, are becoming a dominant approach in robotic control. Benefiting from large-scale multi-modal pre-training, VLAs demonstrate significant advantages in bimanual manipulation [1, 2], long-horizon task planning [1, 61], and mobile manipulation [3]. We adopt DexVLA [2] as our foundational model architecture. Specifically, we employ the Qwen2-VL [62, 6] model as its core VLM. The image encoders project the robot's visual observations into the same embedding space as the language tokens. When handling multiple camera views, the visual embeddings from each view are concatenated. The VLM component produces two types of

outputs: reasoning tokens and action tokens. The action tokens undergo further processing through a projection module composed of two linear layers and a LayerNorm layer. Additionally, we employ the pre-trained 1B ScaleDP[63] module as our action expert. We chose DexVLA because it is among the few open-source VLA models that output unstructured textual reasoning, allowing our approach to effectively harness the VLM's pre-trained knowledge and enabling the VLA model to generalize across diverse scenes.

## 3.2 Model Architecture

**Dynamic mixture-of-expert.** Typically, VLA models utilize a dense vision-language backbone as their foundational architecture. Prior research [7] indicates that multi-modal understanding and robotic manipulation tasks often compete within the parameter space, causing dense VLA models to exhibit erosion of multi-modal comprehension capabilities. To this end, we integrate a Dynamic Mixture-of-Experts (MoE) [64] architecture to effectively handle diverse and complex multi-modal inputs encountered in different tasks. Specifically, our approach utilizes an adaptive routing strategy where expert modules are dynamically selected based on the characteristics of the visual and textual inputs. Ideally, we anticipate that some experts will specialize in task-specific features, such as multi-modal understanding and robot control. These experts focus exclusively on particular tasks, enabling them to learn specialized feature representations through dedicated sets of weights. Conversely, other experts may capture mutually beneficial features shared across multiple tasks, such as spatial reasoning, which is critical for both scene understanding and manipulation. We also expect the gating network to utilize learned criteria to intelligently evaluate input data, selecting the most appropriate subset of experts for activation. This adaptive strategy ensures efficient allocation of computational resources and reduces unnecessary computations. We use the pre-trained MLP weights to initialize the MLP layers for the experts.

*Why static/shared experts are not used?* The key to enabling VLA models to generalize in open-world robotic manipulation lies in preserving the pre-trained knowledge. For architectures like Qwen2-VL — whose LLM component lacks native MoE support — introducing static or shared experts would disrupt the original model structure. Such architectural alterations risk rapidly degrading the VLM's pre-trained knowledge, compromising its reasoning capabilities. Dynamic MoE circumvents this issue by preserving the LLM's intact architecture while selectively activating expert modules. This approach ensures the foundational knowledge remains undisturbed while enabling task-specific adaptation. Our empirical studies in Table3 confirm that dynamic MoE is critical for maintaining the open-world reasoning necessary for generalizable manipulation, as it balances knowledge retention with adaptive learning. In practice, we utilize a total of eight experts and dynamically select two experts during inference.

**Reasoning following enhancement module.** A distinctive feature of our method is that the model not only follows given instructions but also aligns robotic actions closely with the generated reasoning. Prior approaches, such as DiffusionVLA [61] and DexVLA [2], utilize FiLM layers to incorporate reasoning tokens. These methods primarily handle in-domain reasoning scenarios typically encountered during training, making FiLM layers sufficient for reasoning alignment. In contrast, our approach deals with diverse, novel reasoning types not encountered in the training data. Therefore, our method requires a more robust and flexible VLA model capable of effectively following complex, out-of-distribution reasoning.

We introduce an enhanced reasoning-following module designed to improve reasoning capabilities in action models. Specifically, we replace the original observation embedding with reasoning tokens projected through MLP. This reasoning representation is then combined with the current timestep embeddings and used to condition the generation of scale and shift parameters, effectively injecting reasoning context into the model. Importantly, we incorporate this mechanism exclusively into the latter half layers, rather than uniformly across all layers. This design choice aligns with findings from prior studies, such as PointVLA [65] and GR00T N1 [66], which suggest that modifications to the deeper layers of action experts have a smaller impact on robot control. Our results demonstrate that this selective integration allows the model to robustly handle open-world reasoning scenarios without sacrificing in-domain accuracy.

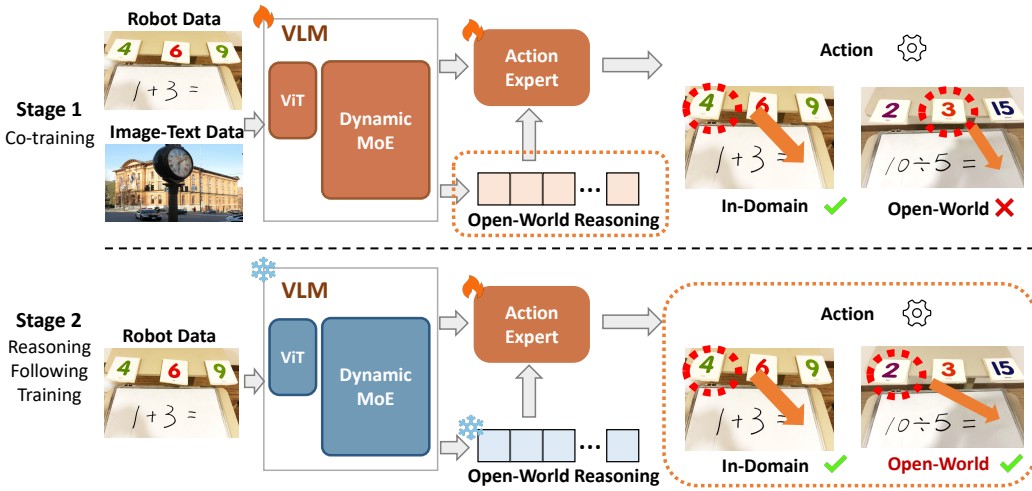

Figure 3: **Training Strategy.** We leverage a two-stage training strategy. In the first stage, we perform co-training on image-text data and robot data to empower VLA with open-world reasoning capabilities. In the second stage, we freeze the entire VLM and train only the action expert, thereby preserving open-world reasoning while enhancing instruction-following abilities in VLA.

## 3.3 Training Strategy

Our previous section introduced the neural architecture of ChatVLA-2, which primarily focuses on enabling the VLA model to more effectively extract common knowledge from pre-trained data and robot actions, guiding the robot to adhere more closely to the generated reasoning. However, we argue that this alone is insufficient for effectively training a general-purpose VLA model. Specifically, mixing image-text data and robot data during training makes it challenging to control the learning process effectively. To address this, we propose a dual-stage training strategy designed to enhance the smoothness of robotic control and increase the success rate of task completion.

**Empowering VLA with open-world embodied reasoning and understanding.** Co-training on image-text and robot data is essential for enabling the robot foundation model to reason and understand scenes in the wild. During this stage, we train the model on both tasks, specifically using datasets COCO [67], TextVQA [68], and GQA [69]. We also construct a dataset of image-text pairs involving robotics scenarios for fine-tuning purposes. Additional details are provided in the Appendix. We apply text augmentation techniques to increase query diversity across all training data. We deliberately avoid selecting training data to bias the VLA toward specific skills such as OCR, mathematical reasoning, or spatial reasoning, as our goal is to utilize pre-trained knowledge for open-world manipulation.

For robot data, we collect 600 trajectories from a math-matching game and 300 trajectories from a toy placement experiment. Similar to DexVLA and $\pi_{0.5}$, all robot data are annotated with reasoning phrases. We maintain an image-text data to robot data ratio of 1:3. This setup follows previous methods. The model undergoes training for 50k steps, beginning with an initial learning rate of 2e-5 and a warm-up phase for the first 3k steps. Subsequently, we apply a cosine learning rate scheduler, scaling down the learning rate to 2e-6.

**Enhancing reasoning-following in VLA.** By jointly training the model on both image-text data and robot data, it learns to reason, recognize, and effectively act within open-world scenarios. The initial stage preserves a significant portion of the pretrained knowledge. However, since our method aims for robots to perform tasks in open-world environments, the reasoning required may not be presented in the training data. Thus, it becomes particularly crucial to strengthen the connection between reasoning and action, ensuring that actions accurately follow and execute the reasoning outcomes for generalizable robot control.

Specifically, we freeze the pretrained VLM and only train the action expert. By keeping the VLM fixed, we effectively preserve the pretrained knowledge acquired in the initial training stage. Consequently, the robot's actions are guided not just by the initial language instructions and image observations but

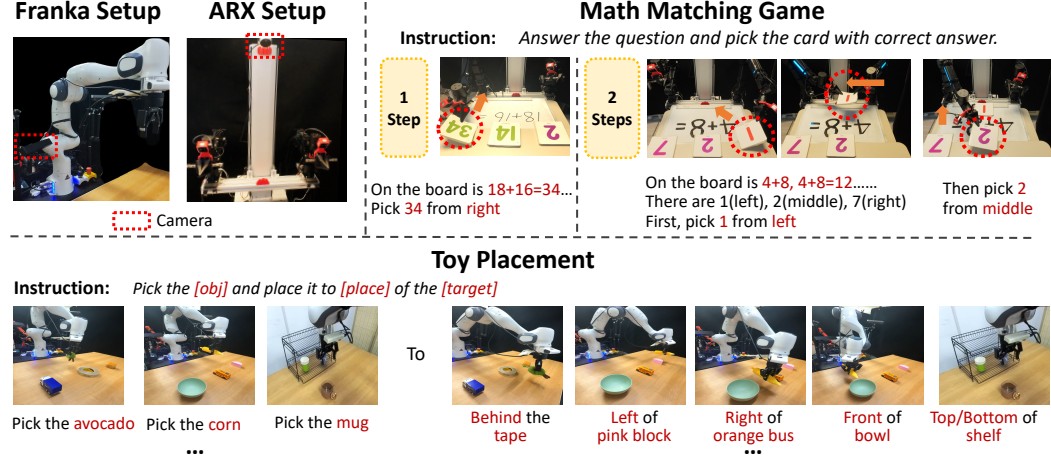

Figure 4: **Experimental setup for math matching game and toy placement.** We use a Franka Emika robot equipped with a Robotiq gripper to pick and place items at specified target locations. We utilize the ARX R5 bimanual robots with a top camera of RealSense L515. Our experiments demonstrate that the proposed method successfully completes tasks involving previously unseen spatial instructions and novel objects.

also significantly by the reasoning outputs generated by the upper layers of the model. We found this strategy particularly beneficial in enhancing the model's understanding and responsiveness to previously unseen reasoning scenarios.

## 4 Experiments

In this section, we conduct extensive real-robot experiments to demonstrate that the end-to-end model is capable of open-world reasoning and understanding and can effectively transfer this knowledge to interactions with the physical world. We do not evaluate using simulation benchmarks, as the VLA capabilities demonstrated by our approach exceed what current simulation benchmarks can assess.

We specifically select two scenarios—math matching games and toy placement task to comprehensively evaluate our proposed method. These experiments examine the model's proficiency in mathematical reasoning, spatial reasoning, optical character recognition (OCR), and object recognition and localization, most within an open-world context involving scenarios that were not part of the training dataset.

### 4.1 Mathematical Reasoning: Math Matching Game

**Evaluation metrics.** We report three types of metrics to evaluate the ability of ChatVLA-2 in manipulation, reasoning, and understanding in both in-domain and open-world. 1) Manipulation success rate: We report the average success rate to measure whether the model completes the task or not. 2) OCR: For OCR, we assign 1 point for correctly recognizing hand-written numbers, 1 point for identifying card values and their positions and 2 points for correctly recognizing the sign. 3) Mathematical reasoning: For mathematical reasoning, we assign 1 point for a correct answer and 1 point for correctly selecting the card.

**Experimental setup.** We consider both in-domain and open-world settings. Specifically, for the in-domain evaluation, all numbers and mathematical symbols exactly match those in the training dataset. However, since numbers and symbols are handwritten, variations in calligraphic style inevitably occur. For the open-vocabulary setting, the mathematical equations tested are entirely absent from the training data.

**Robot setup.** We utilize the bimanual, ALOHA-style robot arm system, ARX-R5, featuring two arms, each with 6 degrees of freedom (6-DoF) and equipped with a top RealSense L515 camera.

This configuration results in a 14-dimensional combined state and action space. Data collection is performed through teleoperation equipment at a frequency of 50 Hz.

**Experimental results.** The experimental results are presented in Table 1. We compare our method against several state-of-the-art models, including Octo [70], Diffusion Policy [32], OpenVLA [10], GR00T N1 [66], DexVLA [2], ChatVLA [7], and $\pi_0$ [1]. We first examine the in-domain performance. For mathematical reasoning and OCR tasks, only a few models such as DexVLA and ChatVLA can output language-based responses. They demonstrate reasonable accuracy in reasoning and OCR tasks, achieving performance comparable to ChatVLA-2. Similarly, in manipulation tasks, ChatVLA-2 does not significantly outperform models like $\pi_0$ and DexVLA, which already exhibit near-perfect performance.

However, substantial differences emerge in open-world scenarios. Even ChatVLA, despite its multimodal understanding capability, fails these tasks when the robot control expert is activated. Consequently, none of the compared methods successfully completed any manipulation tasks in open-world conditions. In contrast, ChatVLA-2 achieves meaningful performance: **3.58 in OCR accuracy, 1.73 in mathematical reasoning accuracy, and 82.7% manipulation success rate**. These experiments highlight the core contribution of our approach: although it may not significantly outperform others in well-trained (in-domain) manipulation tasks, ChatVLA-2 demonstrates substantial superiority in open-world scenarios, successfully handling novel mathematical equations and unfamiliar typography. This represents a significant advancement from zero to effective generalization capability.

## 4.2 Spatial Reasoning: Toy Placement

**Evaluation metrics.** We measure the model with three metrics. First of all, similar to the previous experiment, we report the average success rate of robot action success. Additionally, we provide the open-world object recognition performance in the reasoning process. In the output reasoning, the model needs to output the bounding boxes for the objects that are targeted.

**Experimental setup.** We consider both in-domain and open-world settings. For in-domain evaluation, all objects appear in the training set. For open-world evaluation, the target and reference objects are entirely unseen during training. The model must recognize all objects in an open-world setting, identify the reference objects mentioned in the instruction, understand spatial relations, and execute the placement accordingly.

**Robot setup.** We utilize a 7-Degree-of-Freedom Franka Emika robot equipped with a Robotiq gripper. We use one ZED 2 camera positioned on the right side. Data collection is performed using teleoperation equipment at a frequency of 15 Hz.

**Experimental results.** The experimental results are presented in Table 2. In the in-domain setting, our proposed method performing comparably to DexVLA and $\pi_0$. While ChatVLA was capable of recognizing novel objects in the open-world setting, its performance remained much lower than our method's 0.94. For action execution, models other than our method and $\pi_0$ exhibited near-random success rates in this setting. Even ChatVLA, despite demonstrating some reasoning ability, showed limited open-world robot manipulation ability. In contrast, our method achieved an average success rate of 81.4%, representing a 3.52-times improvement over DexVLA. This result highlights strong spatial reasoning capabilities and reasoning-following capabilities of our method in open-world scenarios.

## 4.3 Ablation Study

**How important is mixture-of-expert in VLA?** This section investigates whether the mixture-of-experts (MoE) mechanism in VLA is crucial for enabling VLA models to generalize for reasoning and understanding in an open-world setting. Specifically, using the exact same training configuration, we compare the baseline models that do not incorporate MoE. Since MoE introduces additional computational overhead during inference, we further compare the model with a larger VLA configuration, specifically the 7B VLM, which has a significantly higher number of parameters at test time.

The experimental results are presented in Table 3. We conducted experiments on the math matching game and observed a significant drop in the average success rate. We hypothesize that this decline is due to conflicts in the parameter space between robotic actions and reasoning/understanding.

Table 1: **Results on the math matching game.** We evaluate multiple models on both in-domain settings, where the data is presented in the training data, and open-world setups. We evaluate average score of **OCR (4 scores in total)** and **mathematical reasoning (2 scores in total)**, and average success rate of task execution at both setups.

| Method | In Domain | | Open-World | | |
| | Reasoning Score | Success Rate | OCR Score | Math Reasoning Score | Success Rate |
|---|---|---|---|---|---|
| Octo [70] | / | 2/13 | / | / | 0/52 |
| Diffusion Policy [32] | / | 7/13 | / | / | 3/52 |
| OpenVLA [31] | / | 2/13 | / | / | 0/52 |
| GR00T N1 [66] | / | 4/13 | / | / | 3/52 |
| DexVLA [2] | 5.2/6 | **12/13** | 0.21/4 | 0.06/2 | 10/52 |
| ChatVLA [7] | 5.8/6 | 10/13 | 1.08/4 | 0.42/2 | 4/52 |
| $\pi_0$ [1] | / | **12/13** | / | / | 8/52 |
| **ChatVLA-2 (Ours)** | **6.0/6** | 11/13 | **3.58/4** | **1.73/2** | **43/52** |

Table 2: **Results on the toy placement task.** We evaluate multiple models on both in-domain settings, where the data is presented in the training data, and open-world setups. We evaluate average object recognition score, spatial affordance score and task success rate at both setups.

| Method Manipulation | In Domain | | | Open-World | | |
| | Object recognition | Spatial Affordance | Avg. Success Rate | Object recognition | Spatial Affordance | Avg. Success Rate |
|---|---|---|---|---|---|---|
| Octo [70] | / | / | 19/67 | / | / | 13/156 |
| Diffusion Policy [32] | / | / | 52/67 | / | / | 17/156 |
| OpenVLA [10] | / | / | 23/67 | / | / | 10/156 |
| GR00T N1 [66] | / | / | 31/67 | / | / | 12/156 |
| DexVLA [2] | 1 | 0.97 | **63/67** | 0.23 | 0.12 | 36/156 |
| ChatVLA [7] | 1 | 0.97 | 60/67 | 0.71 | 0.35 | 22/156 |
| $\pi_0$ [1] | / | / | 61/67 | / | / | 25/156 |
| **ChatVLA-2 (Ours)** | 1 | **0.99** | 61/67 | **0.94** | **0.88** | **127/156** |

The mixture-of-experts approach effectively disentangles the feature spaces associated with these conflicting features. Furthermore, we find that increasing the number of parameters to 7B does not alleviate these conflicts. Upon investigating the cause of the failure, we discovered that for unseen mathematical equations, both dense models fail completely. By examining the mathematical reasoning and OCR scores, we find that when the dense models encounter unseen equations, they often fail to arrive at the correct answer and, in most cases, recognize the wrong answer instead.

**Ablation study on two-stage training.** Our paper proposes a two-stage training strategy designed explicitly to enable VLA models to act effectively in open-world scenarios and consistently follow generated reasoning. Table 4 presents the ablation study isolating the effects of Stage 1 and Stage 2 on model performance in the math matching game. When Stage 2 was excluded, the model's robotic control performance in open-world scenarios dropped to 23% under the same number of training steps. This suggests that while open-world reasoning is generated in Stage 1, it has not been effectively injected in action execution. In contrast, removing Stage 1 resulted in a near-zero score of open-world reasoning capabilities, including both OCR and mathematical tasks, which highlights the critical role of co-training with image-text data.

### 4.4 Results on Multimodal Understanding and Visual-Question Answering

We have conducted extensive evaluations across 12 diverse multi-modal understanding benchmarks, covering tasks such as document understanding (DocVQA), chart and scientific reasoning (ChartQA, AI2D), OCR-based question answering (TextVQA, OCRBench), and real-world fine-grained recognition (InfoVQA, RealWorldQA, MMStar). We also present the results of baseline model ChatVLA, as is shown in Table 5.

Table 3: **Ablation on mixture-of-expert.**

| Method | OCR | Math | Avg. |
|---|---|---|---|
| **Dynamic MoE** | **3.58** | **1.73** | **43/52** |
| Static MoE + Dynamic MoE | 2.38/4 | 0.92/2 | 11/52 |
| Shared MoE + Dynamic MoE | 3.07/4 | 1.12/2 | 25/52 |
| 3B Dense Model | 0.04 | 0.00 | 2/52 |
| 7B Dense Model | 0.08 | 0.00 | 8/52 |

Table 4: **Ablation on training strategy.**

| Stage 1 | Stage 2 | Math Matching Game | | |
| | | OCR | Math | Avg. |
|---|---|---|---|---|
| ✓ | | 3.20 | 1.33 | 12/52 |
| | ✓ | 0.15 | 0.04 | 3/52 |
| ✓ | ✓ | **3.58** | **1.73** | **43/52** |

Table 5: **Understanding task:** Evaluation of VLAs on 7 VQA benchmarks and 5 Multimodal Understanding benchmarks. Boldface denotes top-ranked methods.

| Method | # Params | VQA Benchmarks | | | | | | | Multimodal Understanding Benchmarks | | | | |
|---|---|---|---|---|---|---|---|---|---|---|---|---|---|
| | | TextVQA | DocVQA | InfoVQA | AI2D | ChartQA | MTVQA | RealWorldQA | MMMU | MMStar | MME | OCRBench | HallBench |
| OpenVLA | 7B | 0 | 0 | 0 | 0 | 0 | 0 | 0 | 0 | 0 | 0 | 0 | 0 |
| ECoT | 7B | 0 | 0 | 0 | 0 | 0 | 1.7 | 0 | 5.4 | 0 | 0 | 12 | 0.9 |
| DiVLA | 2B | 7.5 | 15.2 | 14.7 | 43.1 | 17.2 | 6.2 | 25.2 | 17.2 | 21.1 | 186.5 | 294 | 9.0 |
| ChatVLA | 2B | 71.2 | 83.3 | 53.3 | 67.6 | 59.9 | 11.5 | 57.0 | 37.4 | **47.2** | 1435.2 | 729 | 39.9 |
| **ChatVLA-2(Ours)** | 4B | **78.2** | **87.6** | **59.7** | **69.9** | **71.6** | **20.3** | **61.7** | **38.8** | 42.3 | **1464.5** | **792** | **43.9** |

The results demonstrate consistent improvements on 11 out of 12 benchmarks, with particularly notable gains in tasks requiring strong multi-modal understanding capabilities—such as +7.0 on TextVQA, +11.7 on ChartQA, and +6.4 on InfoVQA. These results indicate that key understanding abilities, including fine-grained recognition, OCR, and multimodal reasoning, are retained from the pre-trained VLM.

## 5    Conclusion

Imitation learning typically requires extensive data to master specialized skills for particular tasks. Developing models capable of reasoning and general understanding within open-world scenarios remains a frontier research topic that has yet to be thoroughly explored. In this work, we introduce ChatVLA-2, which endows vision-language-action (VLA) models with the capability to perform diverse tasks by leveraging innate reasoning and understanding abilities derived from pretrained vision-language models in an end-to-end manner. Our core contribution is the introduction of a dynamic Mixture-of-Experts (MoE) module integrated atop a pretrained vision-language backbone. This module efficiently manages different task requirements, where certain experts share common multimodal features, while others are dedicated to task-specific representations. Additionally, we propose a two-stage training strategy: initially, we guide the VLA model to establish connections between pretrained multimodal knowledge and robotic actions; subsequently, we introduce a reasoning-following stage, enabling the model to comprehend reasoning outputs and effectively translate them into corresponding actions.

## Acknowledgments

This work is supported by the Sci-Tech Innovation Initiative by the Science and Technology Commission of Shanghai Municipality (24ZR1419000), and the National Science Foundation of China (12471501).

## Author Contributions

**Project Leader & Advisor**: Yichen Zhu
**Paper Writing**: Yichen Zhu, Zhongyi Zhou
**Algorithm Development**: Zhongyi Zhou, Yichen Zhu, Junjie Wen
**Policies Training & Evaluation**: Zhongyi Zhou, Yichen Zhu
**Data Processing**: Zhongyi Zhou, Xiaoyu Liu
**Leadership**: Yichen Zhu, Chaomin Shen

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

| Table 6: **Ablation study on number of experts.** | | | | Table 7: **Ablation study on reasoning-following enhancement module.** | |
| --- | --- | --- | --- | --- | --- |

| Expert numbers | Top-k numbers | OCR | Math |
| --- | --- | --- | --- |
| 8 | 2 | **3.58** | **1.73** |
| 6 | 3 | 2.42 | 1.26 |
| 4 | 2 | 1.87 | 0.94 |

| Method | Avg. success rate |
| --- | --- |
| **Latter-half-layer injection** | **43/52** |
| Full-layer injection | 36/52 |
| Former-half-layer injection | 22/52 |

## A  Limitation

Our work investigate to retain the pre-trained knowledge from the vision-language model in vision-language-action model. As such, the VLA are able to reasoning over the image observation and language instruction, and enforce the action model to follows such reasoning. Currently, we are unable to fully retain the pre-trained knowledge from VLM. We observe that it is inevitable that many capacity disappear during the fine-tuning with robot data. This is the most challenging part, and current approach cannot fully resolve this problem. We leave this to the future work. Also, our current method is mainly conducted on table top tasks. We aim to expand the embodiment to mobile manipulator to perform more long-horizon and complex real world tasks in the future.

## B  Implementation Details

### B.1  Training details.

We utilize 8 NVIDIA H800 GPUs (80GB each) for training. We adopt mixed-precision training (FP16) and use the AdamW optimizer. For training stage 1, we co-train on image-text data and robot data, setting the initial learning rate to 2e-5 and training for 15k steps. For training stage 2, we freeze the VLM backbone. The model is trained for 50k steps, starting with a learning rate of 2e-5 and a warm-up phase over the first 3k steps. In both stages, we apply a cosine learning rate scheduler, scaling down the learning rate to 2e-6. The total training cost is 340 GPU hours.

### B.2  Data details.

**Image-text data composition.** The image-text dataset used in our experiments integrates samples from multiple established benchmarks, including COCO, TextVQA, and GQA, alongside additional data specifically constructed to align with our task formulation. To ensure balanced representation, we incorporate approximately 32k samples from COCO, 20k from TextVQA, and 54k from GQA. These robotics-related image-text pairs employ the reasoning template used in the toy placement task, as illustrated in B.2. Furthermore, we utilize data from RoboPoint, comprising approximately 2k samples collected within a simulated environment. Although the RoboPoint data exhibits lower visual quality due to visual discrepancies and camera viewpoints, our experiments indicate that including this data enhances the visual-language alignment (VLA) model's spatial understanding capabilities. Additionally, we gathered 5k samples from real-world environments, covering both tabletop setups and broader scenes. These samples follow a similar annotation format to the LLaVA dataset, utilizing a question-answering structure. All collected data is combined and utilized collectively during training in our method.

**Data pre-processing.** For the image-text data, we limit each example to a maximum of 5 dialogue turns. If an instance originally contains more than 5 turns, we retain the first turn and randomly sample four additional turns from the remainder. For the TextVQA dataset, we specifically select samples that do not contain numeric OCR tokens or mathematical operators, as our goal is to utilize pre-trained knowledge for open-world manipulation. We use the image resolution of $320 \times 240$.

**Reasoning templates of robot data.** All our robot data are annotated with sub-reasoning, similar to the approach used in $\pi_{0.5}$ and DexVLA. We initialize these reasoning annotations with fixed templates and then augment them using GPT-4o, following a pipeline analogous to the one employed in training large language models. This method allows us to keep our reasoning phrase flexible, such that the action expert would not dominate by certain template.

# C   More Ablation Studies

We have discussed the importance of some key components in our ChatVLA-2 in the main text, including the choice of mixture-of-experts and the two-stage training strategy. In this section, we will further discuss the following questions:

## C.1   Ablation study on number of experts.

We conduct experiments to check how many experts we should use to better obtain pretrained knowledge from VLM while maintaining appropriate resource consumption. As is shown in Table 6, experimental results indicate that increasing both the total number of experts and the number of experts selected during inference can enhance the model's generalization ability in robotic scenarios.

A possible explanation for this phenomenon is that, a limited number of experts tend to develop selection biases toward visually similar task images in such scenarios. This can lead to overfitting on robot data and result in the neglect of the pretrained VLM knowledge, ultimately degrading performance.

## C.2   Ablation Study on Layers for Injecting Reasoning-Following Enhancement Module.

As shown in the main text, we replace the original observation embedding with reasoning tokens and use them to condition the generation of scale and shift parameters in the latter half layers of the action expert. This mechanism effectively injected reasoning context into the model. In this section, we conduct experiments on the place of injecting reasoning. The results are shown in Table 7.

Experiments show that the former half layers of action expert significantly impacts action generation stability. Introducing reasoning information into the former half layers actually increases instability in the generated actions, which in turn significantly reduces task success rates. We hypothesize that this effect may due to our design choice of replacing the original observation embedding with reasoning information. One possible explanation is that the observations themselves may carry critical information for action generation, and their removal could negatively affect performance.

