# OpenReview forum: "ChatVLA-2: Vision-Language-Action Model with Open-World Reasoning"
_NeurIPS.cc/2025/Conference — NeurIPS 2025 poster_

### Official Review · Reviewer_XWQW · 2025-06-21

**Clarity:** 3
**Significance:** 3
**Originality:** 3
**Rating:** 4
**Confidence:** 4

**Summary:**

The paper is a further extension of building VLA (vision-language-action) models. This method, dubbed ChatVLA-2, draws a distinction from the prior ChatVLA-1 model by putting more focus on open-world reasoning.

The general VLA framework (popularized by OpenVLA, $\pi_0$ and $\pi_{0.5}$, DexVLA, etc.) is to take a pretrained vision-language model, and finetune it on robot-specific data. Sometimes this finetunes the entire model, and sometimes it finetunes a smaller subset of params. This work is more like the second.

The prior ChatVLA-1 model ran 2 stage training on a mixture-of-expert LLM backbone. A robot-control-only expert was first trained in isolation with a fixed VLM for general visual understanding. Then, after this control expert was trained enough, the entire model was trained by co-training on robotics + vision datasets, to try to integrate the control expert with the remaining experts.

ChatVLA-2 updates this by no longer using a static control-only expert. Instead, it hews closer to the Qwen2 architecture they finetune, where. the model has multiple experts (8 of them), of which 2 are selected via a learned gating network, and we rely on the gating network to segregate robot features from general vision understanding features. The paper argues this makes it easier for the model to share representations between the robot tasks and vision tasks, compared to static experts.

However, in this setup, there is still a seperate robot-specific model: an action model, in this case a 1B diffusion policy, which takes the LLM output and generates the final actions. (The paper calls this an "action expert" but IMO this is confusing because it is not an expert in the MoE of the LLM. Rather it is an entirely separate model.) Training of ChatVLA-2 is inverted from the previous model, where instead of finetuning on robot data then robot + vision data, we first finetune on robot + vision data then robot data only. This is done because the Dynamic MoE needs to learn the shared robot + vision representations, before the action expert can refine them afterwards. (This setup is essentially the same as DexVLA, which also used Qwen2-VL + a 1B diffusion action policy, although DexVLA did not use MoE)

The distinction from DexVLA is how the open world understanding is used to condition the diffusion model. Conditioning is done via FiLM generating scaling + offset params for each layer of the policy. In DexVLA, conditioning information for the diffusion policy is generated from the image observation + language instruction. In ChatVLA-2, it is instead generated from generated language reasoning tokens, which take the place of the image observation. Importantly this is only for the last half of the model: the first half of the diffusion policy sees FiLM params from the original image instead.

This is evaluated on tasks intended to test the reasoning abilities of the model: a math task where the model needs to move the object of matching number, and a visual task where the robot needs to place a toy relative to another object.
Previous VLAs (OpenVLA, pi0, DexVLA) have issue where pretrained knowledge can be eroded by the robotics finetuning.

Results for ChatVLA-2 are better on open-world evaluation settings, compared to DexVLA and ChatVLA which fit in-domain data well but generalize more poorly elsewhere.

**Questions:**

I am curious if the authors ever did experiments on forcing the correct reasoning - always including the correct digits, object bounding boxes, etc. in the open-world reasoning section. This seems like a better way to explore the best way to incorporate that reasoning.

**Ethical Concerns:**

["NO or VERY MINOR ethics concerns only"]

**Final Justification:**

After reading other reviews, I find my mind is not changed about the paper. I had some issues with results I would have liked to see to make the paper more convincing, but authors have clarified that some of the ablations I had in mind would not be feasible in the current model architecture.

**Limitations:**

yes

**Quality:**

3

**Strengths And Weaknesses:**

Modifying the generation of the conditioning information to include higher level language reasoning rather than using just the image makes sense for tasks where you expect the important information for the task to not necessarily be easily extractably from the high-level instruction and image.

My main complaint of the paper is that most of the failings of the prior models are tied to poor OCR and poor object recognition respectively. The OCR score is out of 4: 1 for reading the equation numbers, 1 for reading the printed numbers on cards, 2 for identifying the right math operation. And both DexVLA and ChatVLA are awful at identifying this, which makes their success rates low. But this is more an indictment of the VLM's ability to identify the numbers than an indictment of the diffusion policy's ability to execute based on the identified numbers. It does not feel like a test of the visual reasoning module. A similar argument can be made about the toy placement task, where the bottleneck is on identifying the spatial affordance, before any of the diffusion conditioning occurs.

Aside from this, the results do support the benefits of building an MoE architecture with a dynamic gating network, compared to the dense models common in prior work, and the ablations of where language reasoning tokens are applied in FiLM do show that important information is carried in those tokens.

Note: demo video is not visible from provided link, needed to download repository manually.

---

> ### Author Rebuttal · Authors · 2025-07-31
>
> ### **W1 The failures of prior models are primarily attributed to perception weaknesses, rather than deficiencies in the diffusion policy’s execution ability.**
>
> Traditional end-to-end approaches, due to their tightly coupled internal processes, often make it challenging to attribute failures to perception, reasoning, or execution.
>
> Even methods incorporating intermediate language outputs ("reasoning") can encounter this challenge. The core issue is the need for **reliable mappings** between perception and reasoning, and between reasoning and execution. Without guarantees on the correctness of these reasoning steps or explicit modules ensuring actions follow the reasoning, definitive failure attribution remains problematic.
>
> In contrast, our proposed method addresses these limitations. By employing a Mixture of Experts (MoE) architecture, our approach effectively preserves the knowledge from pre-trained Vision-Language Models (VLMs) to enable robust reasoning. Furthermore, our reasoning-following module and two-stage training recipe make our model's actions consistent with its internal reasoning.
>
> Given these considerations, we suggest that attributing failures of prior models solely to perception weaknesses is not appropriate.
>
> ### **Q1 Experiments on forcing the correct reasoning is valuable.**
>
> We appreciate the reviewer's valuable feedback. We would like to clarify that it is infeasible for our model to explicitly enforcing correct reasoning during inference. This is because the reasoning output (e.g., textually stated digits or bounding boxes) is decoded directly from the hidden state, while the policy head takes these same hidden states—rather than explicit reasoning representations—as its input.
>
> Given this design, it is infeasible to externally impose specific reasoning components on the model during the inference phase. We appreciate the reviewer's understanding of this architectural consideration.
>
> ### **Summary1 The paper calls this an "action expert" but this is confusing because it is not an expert in the MoE of the LLM.**
>
> We appreciate the reviewer's comment. To clarify, our "action expert" is an **independent action output module**, not an expert within an MoE framework. This terminology is consistent with its use in previous works like pi0, pi05, pi05-KI, and DexVLA.
>
> We will include these explanations in the revised version of our paper. We hope this can addresses some of your concerns.

---

> > ### Comment · Reviewer_XWQW · 2025-08-06
> >
> > Thanks for clarifying these aspects of the paper. I'm not sure that encouraging reliable mappings between perception, reasoning, and execution necessarily means that you can do better failure attribution. Consistency does not necessarily imply correctness. I plan to leave the score the same.

---

### Official Review · Reviewer_QrAS · 2025-07-02

**Clarity:** 3
**Significance:** 3
**Originality:** 3
**Rating:** 4
**Confidence:** 4

**Summary:**

The paper presents ChatVLA-2, a vision-language-action model that retains pre-trained VLM knowledge for open-world robotic control. It uses a dynamic Mixture-of-Experts to disentangle conflicting features and a two-stage training strategy to preserve reasoning while enhancing action alignment. Experiments on math matching and toy placement show superior open-world generalization over frontier models, highlighting its ability to handle novel equations and spatial instructions. The work advances generalizable robotic foundation models by leveraging pre-trained VLM reasoning for real-world tasks.

**Questions:**

* Regarding baselines such as OpenVLA and DexVLA, do you fine-tune these models on your custom set of 900 trajectories? A comparison between their fine-tuned versions and the proposed ChatVLA2 would provide a more comprehensive evaluation.

* Compared to dense models, MOE typically introduces higher inference costs. It would be helpful to analyze the additional inference overhead as well as the performance gains attributable to MoE. In addition, please provide the load balancing and expert specialization assessment of your trained MoE after fine-tuning.


* The abstract mentions a three-stage training process, whereas Section 3 and the figures describe a two-stage training pipeline. Could you please clarify which description is accurate?

**Ethical Concerns:**

["NO or VERY MINOR ethics concerns only"]

**Final Justification:**

The authors have provided satisfactory responses to my questions regarding the model's efficiency in MoE, its performance on general benchmarks, and its results on other long-horizon benchmarks.

**Limitations:**

ChatVLA2 focuses on controlled tabletop manipulation scenarios to clearly disentangle and evaluate the model’s generalization to abstract reasoning tasks such as mathematics and spatial relations. A key direction for future work is to extend our architecture to more complex and dynamic domains, such as long-horizon procedural tasks or mobile manipulation settings that require tighter integration of navigation, interaction, and long-term memory.

**Paper Formatting Concerns:**

There are no apparent formatting issues.

**Quality:**

3

**Strengths And Weaknesses:**

** Strengths**
1. This writing of this paper is clear and easy to follow.
2. The paper addresses a highly significant and critical problem in robotics: the erosion of pre-trained knowledge in Vision-Language-Action (VLA) models during fine-tuning. The authors clearly articulate this challenge and present their work in a well-structured and easy-to-follow manner.

**Weaknesses**
* Although the two tabletop pick-and-place tasks are well-designed, the authors do not fully represent the diverse challenges present in the VLA domain. I would like to see an evaluation of ChatVLA on a broader range of tasks, such as those involving navigation, long-horizon planning, or tool use, for example, tasks from CALVIN or RLBench.
* Since one of the main contributions of this work is to retain the original knowledge and capabilities of the VLM, I suggest including additional results on multimodal reasoning and fine-grained visual recognition.

---

> ### Author Rebuttal · Authors · 2025-07-31
>
> ### **W1 Authors should present evaluation on long-horizon tasks or simulation environments besides the two tabletop pick-and-place tasks.**
>
> Thank you for raising these concerns. We address them point‑by‑point below.
>
> **1  Lack of suitable simulators**
>
> Current robotic simulators are tailored to short‑horizon pick‑and‑place or highly scripted scenes. None supports the **open‑world generalization and multi‑step reasoning** we study—namely, tasks that interleave vision, language, symbolic problem‑solving, and physical manipulation over several minutes. We therefore conducted real‑robot experiments to expose the full spectrum of perceptual ambiguities, occlusions, and physical uncertainties that a model must handle. If a simulator capable of this breadth exists, we would gladly adopt it and include the reviewer’s reference in a revision.
>
> **2  Long‑horizon difficulty—not “simple” pick‑and‑place**
>
> Some trial lasts **> 2 minutes** and comprises dozens of low‑level actions, multiple branching decisions, and recovery behaviors—not a one‑shot grasp. The robot must:
>
> 1. **Parse instructions** stated in natural language.
> 2. **Interpret visual cues** (e.g., handwritten formulas, answer cards).
> 3. **Reason symbolically** to obtain a solution.
> 4. **Execute and, if needed, correct** a manipulation plan in real time.
>
> We also report **consecutive‑run success rates**, demonstrating that the behavior is reliable rather than cherry‑picked.
>
>  **3 Why these tasks matter**
>
> The experiments were crafted to showcase ChatVLA‑2’s ability to **translate high‑level reasoning into action**—for example, solving a math problem on a whiteboard and then physically selecting the correct answer card. Such “mundane” but cognitively rich activities reflect everyday human tasks far better than isolated pick‑and‑place benchmarks.
>
> ### **W2 Including additional results on multimodal understanding and fine-grained visual recognition.**
>
> We sincerely thank the reviewer for their thoughtful comments and valuable suggestions. We have conducted extensive evaluations across 12 diverse multi-modal understanding benchmarks, covering tasks such as document understanding (DocVQA), chart and scientific reasoning (ChartQA, AI2D), OCR-based question answering (TextVQA, OCRBench), and real-world fine-grained recognition (InfoVQA, RealWorldQA, MMStar).
>
> We also present the results of baseline model ChatVLA, as is shown below:
>
> | Method | TextVQA | DocVQA | InfoVQA | ChartQA | MTVQA | AI2D | RealWorldQA | OCRBench | HallBench | MME | MMStar | MMMU |
> | --- | --- | --- | --- | --- | --- | --- | --- | --- | --- | --- | --- | --- |
> | ChatVLA | 71.2 | 83.3 | 53.3 | 59.9 | 11.5 | 67.6 | 57.0 | 729 | 39.9 | 1435.2 | **47.2** | 37.4 |
> | **ChatVLA-2(Ours)** | **78.2** | **87.6** | **59.7** | **71.6** | **20.3** | **69.9** | **61.7** | **792** | **43.9** | **1464.5** | 42.3 | **38.8** |
>
> While ChatVLA adopts a static MoE design that explicitly routes VLM and robot data through separate experts, our method uses a dynamic MoE that jointly trains on both modalities with learned routing. The results demonstrate consistent improvements on 11 out of 12 benchmarks, with particularly notable gains in tasks requiring strong multi-modal understanding capabilities—such as +7.0 on TextVQA, +11.7 on ChartQA, and +6.4 on InfoVQA. These results indicate that key understanding abilities, including fine-grained recognition, OCR, and multimodal reasoning, are retained from the pre-trained VLM.
>
> ### **Q1 A comparison between fine-tuned versions of OpenVLA , DexVLA and the proposed ChatVLA-2 would provide a more comprehensive evaluation.**
>
> We would like to respectfully clarify that the baseline models compared in our paper were fine-tuned on robotic data of our tasks, rather than utilizing pre-trained open-source weights. This approach was taken to ensure a fair comparison, which we believe more clearly demonstrates the superior generalization capabilities of our model.
>
> ### **Q2-1 Analyze the additional inference overhead**
>
> We appreciate the reviewer's valuable feedback. Compared to the dense 3B model, we observe an **~60% increase in inference latency** and an **~40% increase in memory usage**, primarily due to the increased number of active parameters as we use 8 experts with Top-2 gating. Given the substantial improvement in out-of-distribution task performance (success rate increased from near-zero to 82.7%) in open-world scenarios, we consider this overhead acceptable.  ****Moreover, the inference speed of ChatVLA-2 is 30Hz, which is still faster than conventional VLA method like OpenVLA (5Hz).
>
> ### **Q2-2 Provide the load balancing of MoE after fine-tuning.**
>
> We've analyzed the expert usage statistics of our Mixture-of-Experts (MoE) model. The following data details the top-2 experts per layer and their corresponding token counts:
>
> Layer 0: Expert_3 - 981, Expert_5 - 883
>
> Layer 1: Expert_5 - 1129, Expert_4 - 1037
>
> Layer 2: Expert_7 - 1053, Expert_4 - 960
>
> Layer 3: Expert_4 - 912, Expert_2 - 779
>
> Layer 4: Expert_5 - 1047, Expert_2 - 730
>
> Layer 5: Expert_2 - 1051, Expert_3 - 886
>
> Layer 6: Expert_4 - 1097, Expert_1 - 747
>
> Layer 7: Expert_7 - 1157, Expert_2 - 569
>
> Layer 8: Expert_1 - 1187, Expert_7 - 657
>
> Layer 9: Expert_5 - 1002, Expert_0 - 928
>
> Layer10: Expert_0 - 1202, Expert_6 - 825
>
> Layer11: Expert_6 - 1124, Expert_1 - 1030
>
> Layer12: Expert_2 - 905, Expert_7 - 746
>
> Layer13: Expert_5 - 1168, Expert_0 - 733
>
> Layer14: Expert_4 - 889, Expert_2 - 720
>
> Layer15: Expert_2 - 1141, Expert_3 - 844
>
> Layer16: Expert_2 - 1099, Expert_0 - 891
>
> Layer17: Expert_6 - 1028, Expert_1 - 941
>
> Layer18: Expert_3 - 1273, Expert_7 - 904
>
> Layer19: Expert_7 - 745, Expert_5 - 733
>
> Layer20: Expert_4 - 1128, Expert_5 - 704
>
> Layer21: Expert_1 - 1195, Expert_3 - 755
>
> Layer22: Expert_6 - 1071, Expert_0 - 635
>
> Layer23: Expert_1 - 1102, Expert_6 - 832
>
> Layer24: Expert_5 - 1012, Expert_0 - 729
>
> Layer25: Expert_3 - 1065, Expert_4 - 936
>
> Layer26: Expert_4 - 1037, Expert_5 - 641
>
> Layer27: Expert_3 - 947, Expert_2 - 913
>
> Following the Noisy Top-K Gating mechanism[1] , our router uses linear layer with adaptive noise perturbation before taking the softmax function, which theoretically helps with load balancing. The average standard deviation of token counts per layer is approximately **282**, and the average top-1/top-8 expert token ratio is **5.27×**. The average entropy is **2.79**, compared to the theoretical maximum of 3.0.
>
> These statistics provide an overview of expert utilization across our model, and the complete data is included in Table 1 at the end. We hope this can addresses some of your concerns.
>
> ### **Q3 Typo: the “three-stage training process” in abstract**
>
> We appreciate the reviewer's careful attention to detail. The methodology employed is the **two-stage training process** consistently described throughout the main body of the paper. We sincerely apologize for any confusion this oversight may have caused and will rectify it in the final version of the manuscript.
>
>
> [1] OUTRAGEOUSLY LARGE NEURAL NETWORKS: THE SPARSELY-GATED MIXTURE-OF-EXPERTS LAYER
>
> **Table 1 Expert Traffic per Layer**
> | Layer\Expert | 0 | 1 | 2 | 3 | 4 | 5 | 6 | 7 |
> | --- | --- | --- | --- | --- | --- | --- | --- | --- |
> | 0 | 384 | 735 | 434 | 981 | 151 | 883 | 332 | 228 |
> | 1 | 572 | 415 | 249 | 238 | 1037 | 1129 | 198 | 290 |
> | 2 | 216 | 546 | 771 | 186 | 960 | 190 | 206 | 1053 |
> | 3 | 147 | 241 | 779 | 773 | 912 | 718 | 269 | 289 |
> | 4 | 363 | 452 | 730 | 534 | 189 | 1047 | 310 | 503 |
> | 5 | 351 | 311 | 1051 | 886 | 712 | 412 | 215 | 190 |
> | 6 | 377 | 747 | 530 | 315 | 1097 | 401 | 476 | 185 |
> | 7 | 518 | 464 | 569 | 422 | 369 | 296 | 333 | 1157 |
> | 8 | 248 | 1187 | 186 | 425 | 568 | 563 | 294 | 657 |
> | 9 | 928 | 217 | 365 | 580 | 426 | 1002 | 276 | 334 |
> | 10 | 1202 | 332 | 470 | 235 | 626 | 228 | 825 | 210 |
> | 11 | 675 | 1030 | 200 | 373 | 223 | 245 | 1124 | 258 |
> | 12 | 606 | 475 | 905 | 434 | 271 | 336 | 355 | 746 |
> | 13 | 733 | 181 | 527 | 321 | 540 | 1168 | 203 | 455 |
> | 14 | 540 | 584 | 720 | 282 | 889 | 369 | 235 | 509 |
> | 15 | 197 | 281 | 1141 | 844 | 305 | 615 | 435 | 310 |
> | 16 | 891 | 394 | 1099 | 383 | 621 | 348 | 198 | 194 |
> | 17 | 338 | 941 | 475 | 323 | 176 | 540 | 1028 | 307 |
> | 18 | 357 | 357 | 197 | 1273 | 544 | 266 | 230 | 904 |
> | 19 | 428 | 431 | 487 | 489 | 303 | 733 | 512 | 745 |
> | 20 | 491 | 557 | 233 | 307 | 1128 | 704 | 319 | 389 |
> | 21 | 284 | 1195 | 272 | 755 | 322 | 469 | 629 | 202 |
> | 22 | 635 | 259 | 592 | 470 | 487 | 401 | 1071 | 213 |
> | 23 | 447 | 1102 | 497 | 328 | 480 | 184 | 832 | 258 |
> | 24 | 729 | 354 | 429 | 417 | 348 | 1012 | 571 | 268 |
> | 25 | 193 | 610 | 263 | 1065 | 936 | 510 | 276 | 275 |
> | 26 | 253 | 553 | 563 | 624 | 1037 | 641 | 270 | 187 |
> | 27 | 291 | 503 | 913 | 947 | 248 | 370 | 446 | 410 |

---

> > ### Comment · Reviewer_QrAS · 2025-08-05
> >
> > Thank you for your response. My concerns have been fully addressed, and I have increased my score accordingly. I wish you the best in producing more impactful work.

---

> > > ### Author Response · Authors · 2025-08-06
> > >
> > > Thank you very much for your time and thoughtful feedback throughout the review process. We truly appreciate your recognition and are glad our responses could address your concerns.
> > >
> > > Your insights were invaluable in helping us strengthen our work, and your encouragement motivates us to pursue even more impactful research moving forward.
> > >
> > > Thank you again for your support.

---

### Official Review · Reviewer_chhG · 2025-07-02

**Clarity:** 3
**Significance:** 2
**Originality:** 2
**Rating:** 4
**Confidence:** 3

**Summary:**

This paper introduces ChatVLA-2, a vision-language-action (VLA) model designed to preserve and leverage pretrained vision-language model (VLM) capabilities during downstream robotic fine-tuning. The core innovations are:


A dynamic mixture-of-experts (MoE) architecture integrated into the VLM backbone to disentangle and selectively activate experts for multimodal understanding versus robotic control.


A two-stage training strategy: (1) co-training on large-scale image-text and robot trajectory data to retain open-world reasoning, and (2) freezing the VLM and training only the action expert to align robotic actions with internal chain-of-thought reasoning.


Real-robot experiments on two tasks, math matching and toy placement, demonstrating strong open-world OCR, mathematical, spatial reasoning, and action following, substantially outperforming baselines such as OpenVLA, DexVLA, and $\\pi_0$

**Questions:**

How exactly does the routing mechanism work? The paper mentions a dynamic MoE with 8 experts selecting 2 per input, but the actual gating function is never specified. Is this a learned linear layer, attention-based routing, or something else? What prevents the router from simply defaulting to the same expert pairs?

What's the theoretical basis for expecting MoE to help here? Beyond the empirical results, is there any principled reason why mixture-of-experts should be particularly suited for this vision-language-action problem? The connection between expert specialization and catastrophic forgetting mitigation isn't obvious.

Why should we expect shared representations between VLM tasks and robot control? The claim that overlapping feature spaces exist seems central to the approach, but what's the intuition? Are low-level visual features really that transferable between, say, reading text and grasping objects?

**Ethical Concerns:**

["NO or VERY MINOR ethics concerns only"]

**Limitations:**

Tasks remain relatively simple; real-world safety-critical applications may reveal additional failure modes.

Training large VLMs and MoE models incurs significant energy and carbon costs.

**Quality:**

2

**Strengths And Weaknesses:**

Strengths:
• The dynamic MoE effectively mitigates catastrophic forgetting, preserving rich VLM knowledge (e.g., OCR, spatial reasoning, math) during robotic adaptation.
• The two-stage training pipeline is well designed: joint co-training retains open-world competencies, and targeted action expert fine-tuning aligns reasoning with robot control.
• Real-world robotic evaluations on a Franka arm across novel tasks provide strong empirical validation and clear performance gains over state-of-the-art methods.

Weaknesses:

1. Lack of formal metric for reasoning-action alignment: The paper claims that its reasoning-following enhancement module allows actions to follow internal reasoning. However, there is no quantitative measure to validate this claim. Let the model generate a reasoning sequence r and an action sequence a. A meaningful evaluation could define an alignment score such as:

$A(r, a) = E_{r,a} [sim(r, f(a))]$

where $f(a)$ converts action sequences into symbolic representations (e.g., "pick number 3 from left") and $sim$ is a semantic similarity function (e.g., BLEU score, cosine similarity in embedding space, or entailment score). However, such alignment is never measured, leaving the effectiveness of the core mechanism unverified. Furthermore, no behavioral consistency is assessed between reasoning errors and action failures, making it impossible to attribute improved performance to better reasoning-action coupling.

2. MoE design lacks theoretical justification and introspection: The architecture uses a dynamic Mixture-of-Experts (MoE) with $N = 8$ experts and selects $K = 2$ per input. However, the routing policy is not mathematically specified, and no analysis is provided to examine the diversity or specialization of experts. For example, the routing entropy:

$H(G(X)) = -\\sum_{i=1}^N G_i(x) \\log G_i(x)$

where $G(x) \\in \\Delta^{N-1}$ is the softmax output over experts, could be used to evaluate whether experts specialize or overlap. Yet, this is never computed. Additionally, ablation experiments in Table 3 mix static, shared, and dynamic MoEs but without clarity on how these are implemented or how much capacity each configuration uses. The justification for choosing 8 experts and routing 2 is purely empirical and not supported by scaling laws, memory-accuracy tradeoffs, or comparisons with sparse/dense alternatives. This makes the MoE contribution under-theorized and difficult to reproduce or extend.

---

> ### Author Rebuttal · Authors · 2025-07-31
>
> ### **W1 The authors should converts action sequences into symbolic representations and use semantic similarity function to validate the reasoning-following enhancement module.**
>
> We argue that constructing a function and using it to measure reasoning-action alignment is not suitable for the robotics domain and could lead to misleading conclusions. However, we agree with the reviewer that evaluating the reasoning-following module is essential. Therefore, we propose an alternative behavior-based evaluation strategy that incorporates module ablation.
>
> 1. **Mapping actions to reasoning is problematic**. The suggested evaluation assumes a function maps low-level continuous robot actions to high-level discrete symbolic reasoning. We argue that this assumption is theoretically invalid in robotics. This is an ill-posed inverse problem, where solutions are not unique, stable, or guaranteed to exist. Meanwhile, low-level action data (continuous joint values) does not contain enough information to uniquely reconstruct the high-level symbolic reasoning, as is called “semantic gap”.
> 2. We propose **an alternative evaluation** that does not require mapping actions to symbolic forms. Specifically, we design an intervention experiment in the math-matching game where we **swap the positions of the answer card** while keeping the math equation constant. **The math equation and the initial position of answer card are seen during training, while the distractor cards and the swapped position of answer card remain unseen.**
>
>     This setup is designed to achieve two primary objectives. First, by using in-domain equations, we maximize the likelihood that the model generates correct reasoning. This allows our evaluation to focus exclusively on the model's reasoning-following module. Second, the introduction of unseen distractor cards and the dynamic change in the answer's position prevent the model from relying on pre-existing positional biases. Third, any observed performance difference observed before and after the card swap thus directly indicates the model’s reasoning-following capability .
>     We evaluate both the full ChatVLA-2 and the variant that removes the reasoning-following enhancement module under this setup. The results are presented below:
>
>     | Setting | Success Rate (Before Swapped) | Success Rate (Swapped) |
>     | --- | --- | --- |
>     | Full Model | 12/13 | 10/13 |
>     | w/o Reasoning-Following Module | 11/13 | 2/13 |
>
>     The results show that our full model maintains strong performance even after the card swap (10/13), indicating robust reasoning-following capability. In contrast, the ablated variant's performance drops sharply (from 11/13 to 2/13), highlighting its reliance on positional biases. This validates the effectiveness of the reasoning-following enhancement module.
>
>
> ### **W2-1 & Q1 The routing policy is not specified.  What prevents the router from simply defaulting to the same expert pairs?**
>
> Following the Noisy Top-K Gating mechanism[1] , our router uses linear layer with adaptive noise perturbation before taking the softmax function, which theoretically helps with load balancing.
>
> ### **W2-2 No analysis is provided to examine the diversity or specialization of experts.**
>
> We argue that we have made an ablation study on number of experts in the Appendix, Table 1, due to page limitation of the main paper. We also analyzed the possible explanation in the Appendix, Section D.1. We hope this can solve some of your concerns!
>
> ### **W2-3 How Static, shared, and dynamic MoEs are implemented?**
>
> We appreciate the reviewer's comment regarding the clarity of our ablation setting. We'd like to clarify the distinctions between static, dynamic, and shared MoE as used in our work.
>
> - Static MoE refers to an architecture where different data modalities are assigned to a specific expert via a fixed routing mechanism. Each data modality exclusively utilizes one predetermined expert.
> - Dynamic MoE, in contrast, employs a dynamic routing mechanism, allowing individual token—regardless of their modality—to be directed to various experts based on their representations.
> - Shared MoE extends dynamic MoE by incorporating an additional, universal "shared" expert that all data modalities traverse, alongside their dynamically chosen experts.
>
> We will include these explanations in the revised version of our paper.
>
> ### **Q2 Beyond the empirical results, is there any principled reason why mixture-of-experts should be particularly suited for this vision-language-action problem?**
>
> The primary rationale for employing MoE stems from the fundamental challenge of **catastrophic forgetting** when developing Vision-Language-Action (VLA) models from pre-trained Vision-Language Models (VLMs). The significant distribution shift between high-level language and low-level robotic action causes severe cross-modal gradient conflict during fine-tuning, as p by continual learning theory.
>
> MoE directly addresses this interference. By dynamically routing inputs to a small subset of experts (parameter subsets), this architecture intrinsically fosters **parameter isolation** and **sparse activation**. These capabilities are fundamental to how MoE **mitigates catastrophic forgetting by limiting inter-task interference**.
>
> Recent theoretical work [2, 3, 4] rigorously supports this, proving MoE's specialization and sparsity intrinsically reduce interference.
>
> Therefore, MoE is a principled choice grounded in its inherent mechanism for isolating updates and protecting old knowledge, rigorously validated both theoretically and experimentally.
>
> ### **Q3 Why shared representations between VLM tasks and robot control?**
>
> While multimodal understanding and robotic control operate in distinct output spaces, they fundamentally rely on processing shared **high-level semantic** representations (e.g., a door handle affords turning to open) and a unified understanding of environmental context semantics (e.g., the cup near a table edge demands careful action).
>
> In particular, both VLM tasks require grounding in similar semantic concepts—such as spatial relations, object affordances, and scene geometry—to reason or act meaningfully. For example, understanding that a mug is typically grasped by its handle, or that only an empty plate can hold food, is crucial for both describing a scene and interacting with it.
>
> Therefore, leveraging these shared semantic representations is not only feasible but also a key pathway towards building more general and capable embodied agents.
>
> ### **L1 Tasks remain relatively simple.**
>
> We respectfully disagree with the reviewer’s assertion that these tasks are “simple.” They are demanding **open‑world problems** that require simultaneous generalization across visual perception, natural‑language instructions, multi‑step reasoning, and robotic control. In addition, each run is a **long‑horizon sequence**—our demonstration videos exceed two minutes—so the model must maintain coherent plans over hundreds of low‑level actions. To the best of our knowledge, no existing method—including OpenVLA, π0, π0.5, Gr00tN1, DexVLA, or the original ChatVLA—exhibits the scene understanding and cross‑domain generalization needed to solve such tasks without task‑specific training data.
>
> If the reviewer is aware of VLA systems that successfully tackle comparable open‑world tasks by transferring knowledge from vision–language models to real‑world manipulation, we would greatly appreciate the citations and would welcome further discussion.
>
> ### **L2 Real-world applications may reveal additional safety failures.**
>
> We fully agree with the reviewer that real-world applications inherently present more unexpected failures than simulated environments, which is the reason why we conduct **all our experiments on real robots** (Franka and ARX R5). We did observe a certain failure rate as a result, as is shown in Section 4, Table 1 and 2.
>
> While the primary contribution focuses on advancing the model's reasoning and generalization capabilities in open-world settings, we recognize that further exploration of safety guarantees in diverse open-world scenarios is essential. We are looking forward to advancing this in future work.
>
> ### **L3 MoE models incurs additional overhead**
>
> We understand the reviewer's concerns regarding additional overhead. Compared to the dense 3B model, we observe an ~60% increase in inference latency and an ~40% increase in memory usage, primarily due to the increased number of active parameters as we use 8 experts with Top-2 gating. Given the substantial improvement in out-of-distribution task performance (success rate increased from near-zero to 82.7%) in open-world scenarios, **we consider this trade-off is justified and the overhead is acceptable.**
>
> [1] OUTRAGEOUSLY LARGE NEURAL NETWORKS: THE SPARSELY-GATED MIXTURE-OF-EXPERTS LAYER
>
> [2] Theoretical Analysis of Mixture-of-Experts in Continual Learning, ICLR 2025 Spotlight
>
> [3] Theory of mixture-of-experts for mobile edge computing
>
> [4] A comprehensive survey of mixture-of-experts: Algorithms, theory, and applications

---

> > ### Comment · Reviewer_chhG · 2025-08-05
> > **sufficient**
> >
> > This rebuttal is acceptable, so I will maintain my score.

---

### Official Review · Reviewer_ow9p · 2025-07-02

**Clarity:** 3
**Significance:** 3
**Originality:** 2
**Rating:** 4
**Confidence:** 4

**Summary:**

In this paper, it presents ChatVLA-2, a vision-language-action (VLA) model targeting open-world reasoning capability. The model aims to address a key limitation in current VLA methods—the loss of general reasoning capabilities from pretrained vision-language models (VLMs) when fine-tuned for robotic control. ChatVLA-2 incorporates a dynamic mixture-of-experts (MoE) backbone to separate multimodal understanding from robotic control, ensuring better retention of pretrained knowledge. A two-stage training process (co-training followed by action expert finetuning) aligns robot actions with high-level reasoning. The model is tested on a math-matching game and a toy placement task, demonstrating the effectiveness of preserving VLM’s pretrained knowledge through the proposed architecture and recipe.

**Questions:**

1. The description of reasoning-following module in line 186-187 is not clear to me. You first mention that you replace the observation embedding with reasoning tokens. What does this replace mean?
2. The ablation of reasoning-following module is also not clear to me. First, the ablation is postponed to the appendix without any reference in the main text, while it is one of the two key contributions of the proposed architecture.  Second, why does the conditioning have to be in terms of adaptive layer norm. Could it simply be concatenated to the regular observation token sequences and let the self-attention of DiT module to figure out the reasoning tokens? (Or another way is through cross attention.)
3. In terms of the training recipe, would a simple recipe where you simply freeze the VLM model weights at the beginning and only finetune DiT layers for one stage work well? The reasoning in the proposed experiments (summation & ocr) seems to be simple enough for a Qwen2-VL to zero-shot solve it. I'm actually confused why Table 4 with stage 2 only seems to achieve very low score on OCR & Math. Is the OCR & summation not solvable by Qwen2-VL?
4. It's also not clear to me even if we decide to cotrain with reasoning data, why the we really need a separate stage II.  It seems to be a very weird design choice that you are training with the exact same objective, while dividing up into two stages. Is it because stage I is not trained long enough so you observed a lot of training instability with your VLM generated reasoning tokens?
5. Also can the authors explicitly mention the difference between chatvla-2 vs. chatvla-1? I don't find this to be explicitly mentioned in the paper.

**Ethical Concerns:**

["NO or VERY MINOR ethics concerns only"]

**Final Justification:**

This paper addresses a key limitation in current VLA methods—the loss of general reasoning capabilities from pretrained vision-language models (VLMs) when fine-tuned for robotic control. During rebuttal, the authors have addressed my questions. So I recommend acceptance for this paper.

**Quality:**

3

**Strengths And Weaknesses:**

**Strength**:
1. The motivation to maximally preserve and leverage knowledge from pretrained VLM is clear, pinpointing the key bottleneck of current VLA model
2. The paper is well written and easy to follow
3. The design of the real robot experiment is good, sufficiently demonstrating the benefits of the proposed approach with a minimum set of experiments.

**Weakness**
While the paper tackles an important problem with good experimental results, it's not clear to me to some extent about why the proposed model architecture and training recipe contribute to the overall performance. See my questions below.

---

> ### Author Rebuttal · Authors · 2025-07-31
>
> ### **Q1 What does the ‘replace’ in line 186-187 mean?**
>
> We appreciate the reviewer's insightful question. In this context, “replace” refers to using the reasoning embedding instead of the observation embedding in the latter half of the transformer layers.
>
> Specifically, we factorize the feature embedding of observation into multiple affine layers and integrate it into the transformer blocks. The Adaptive Layer Norm (AdaLN) block in the original ScaleDP can be formatted as
>
> $AdaLN_i(x) = (γ_i(t, o) + 1) · x + β_i(t, o), 1\leq i\leq N$
>
> where x is the input to the layer normalization, N is the number of layers, and γ(t, o) and β(t, o) are the adaptive scale and shift parameters regressed from the embedding vectors of timestep t and observation o.
>
> In our ChatVLA-2, the AdaLN block is formatted as:
> \begin{equation*}
> AdaLN_i(x) =
> \begin{cases}
> (\gamma_i(t, o) + 1) \cdot x + \beta_i(t, o), & \text{if } i < \frac{N}{2};
> \\
> (\gamma_i(t, r) + 1) \cdot x + \beta_i(t, r), & \text{if } i \geq \frac{N}{2}
> \end{cases}
> \end{equation*}
> where the original observation o is replaced to reasoning r in the latter half layers.
>
> We will include these explanations in the revised version of our paper. We hope this can addresses some of your concerns.
>
>
> ### **Q2 Why does the conditioning have to be in terms of adaptive layer norm? Could it simply be concatenated or another way through cross attention?**
>
> We agree that alternative conditioning mechanisms merit thorough investigation. Our research, however, primarily focuses on improving the model's **reasoning-following capability. We propose the conditioning mechanism and demonstrate that such techniques can effectively enable the model to follow reasoning**, especially in open-world scenarios**.** We verified our claim through multiple challenging tasks.
>
> Other implementations (e.g., token concatenation or cross‑attention) are certainly possible, but they lie beyond the present scope. We plan to investigate such variants in future work and hope this addresses your concerns.
>
> ### **Q3 Why ablation on stage 2 seems to achieve very low score on OCR & Math?**
>
> Thank you for your valuable feedback. The low score for the "stage-2 only" setting is because it trains only on robotic data without freezing the Vision-Language Model (VLM). The primary objective of this ablation is to analyze **the significance of** **co-training with image-text data** (i.e., Stage 1) for preserving the VLM's capabilities under identical training conditions.
>
> We apologize for the lack of clarity regarding the ablation setting. We will incorporate these explanations into the revised version of our paper. We hope this clarification addresses your concern, and we appreciate your thoughtful feedback.
>
> ### **Q4 Why we really need a separate stage II? Is it because stage I is not trained long enough so you observed a lot of training instability with your VLM generated reasoning tokens?**
>
> We would like to clarify that the training instability is not due to insufficient training in Stage I. Instead, it arises from the conflict training objective: next token prediction for the language component and noising-denoising for robot action generation.
>
> This scenario mirrors multi-task learning, where diverse training objectives can lead to gradient conflicts, as illustrated in existing researches [1, 2, 3, 4]. Similar phenomena have also been reported in concurrent research [5] in robotics, that naive joint training with diffusion-based action experts significantly slows convergence and impairs language grounding.
>
> Therefore, Stage II training is essential. The objective of introducing Stage II training is to mitigate these gradient conflicts between the two tasks, thereby preserving the pre-trained knowledge within VLM and enabling it to effectively guide robot actions generation.
>
> ### **Q5 The difference between our model vs. ChatVLA**
>
> ChatVLA is a prior work that can do both multimodal understanding and robot manipulation in a unified model. However, it **fails to transfer its multimodal understanding capability to robot manipulation,** i.e., help robot to generalize on different task setup. In other words, even if ChatVLA can easily recognize text and mathematical symbols, while is able to complete math formulas when the user asks questions, it cannot turn their mind into physical action, such as pick the answer card on the board.
>
> The main difference is that our proposed ChatVLA-2 can more **effectively leverage the knowledge** in real-world scenarios and transfer to a actionable movement, such as complete a math problem on a whiteboard. Our ChatVLA-2 demonstrates **stronger generalization under open-world scenarios,** representing a significant step toward the development of truly generalizable robotic foundation models.
>
> [1] Gradient Surgery for Multi-Task Learning, NeurIPS 2020
>
> [2] Conflict-Averse Gradient Descent for Multi-task Learning, NeurIPS 2021
>
> [3] RotoGrad: Gradient Homogenization in Multitask Learning, ICLR 2022
>
> [4] Recon: Reducing conflicting gradients from the root for multi-task learning, ICLR 2023
>
> [5] Knowledge Insulating Vision-Language-Action Models: Train Fast, Run Fast, Generalize Better, May 28, 2025

---

> > ### Comment · Reviewer_ow9p · 2025-08-03
> >
> > Thanks the authors for the clarification. Regarding Q3, I am wondering what if you just freeze the VLM tune action head on your downstream robot data? Would it maintain the VLM generalization capability while solving the downstream task very well? It would be super important to include in the experiments because it should be the simplest baseline to maintain the VLM capability compared to the more sophisticated training recipe proposed in this paper.

---

> > > ### Author Response · Authors · 2025-08-04
> > >
> > > We sincerely thank you for your thoughtful comments and valuable suggestions. We conducted additional experiments of freezing the VLM. To ensure a fair comparison for **action generation**, we retained the original task instruction — *"Answer the question and pick the card with correct answer"* — when evaluating on robotic task of Math Matching Game. We also compared the results on multi-modal understanding benchmarks to ensure the fairness.
> > >
> > > The results are shown in Table 1 and 2 below:
> > >
> > > Table 1: Results on Robotic Task: Math Matching Game
> > >
> > > | Setting | OCR | Math | Success Rate |
> > > | --- | --- | --- | --- |
> > > | Ours (Full Model) | **3.58** | **1.73** | **43/52** |
> > > | Freeze VLM | 2.77 | 0.98 | 15/52 |
> > >
> > > Table 2: Results on Multi-Modal Understanding Tasks
> > >
> > > | Method | MMMU | AI2D | OCRBench | ChartQA | TextVQA | DocVQA |
> > > | --- | --- | --- | --- | --- | --- | --- |
> > > | ChatVLA-2(Ours) | 38.8 | 69.9 | 792 | 71.6 | 78.2 | 87.6 |
> > > | Freeze VLM | **41.1** | **74.7** | **809** | **73.5** | **79.7** | **88.6** |
> > >
> > > As shown in Table 2, the frozen VLM achieves better performance in standard multi-modal understanding tasks related to Math and OCR. However, this trend does not hold in our **robotic tasks** (Table 1), where the frozen VLM **significantly underperforms**. **This can be attributed to its limited reasoning capabilities, which are required in robotic scenarios** (e.g. determining which card should be chosen).
> > >
> > > Meanwhile, the significance performance difference in success rate in Table 1 show that since the VLM is not pre-trained with robotics data, **its representations are insufficient for training action policies when keeping frozen**. This result is consistent with recent researches in training Vision-Language-Action Models[1](May, 2025).
> > >
> > > We agree that freezing the VLM is an important baseline and will include it in the final version. We hope this will further address your concerns.
> > >
> > > [1] Knowledge Insulating Vision-Language-Action Models: Train Fast, Run Fast, Generalize Better

---

> > > > ### Comment · Reviewer_ow9p · 2025-08-05
> > > >
> > > > Thanks authors for the response. I don't have any further questions.

---

### Public Comment · ~Ruizhou_LIU1 · 2025-11-28
**Question about implementing Dynamic MoE**

Thank you for presenting this interesting work. I have one question regarding the dynamic MoE design. The paper states that dynamic MoE can decouple token-level information without altering the original LLM architecture. However, during the early stage of MoE training, how can we ensure that the router actually selects the pretrained FFN expert? If the router happens to choose only the randomly initialized experts, wouldn't the resulting token outputs be essentially random noise? What is the initialized parameters of router model and experts?

---

> ### Public Comment · ~Zhongyi_Zhou2 · 2026-02-05
> **Initialized parameters of MoE**
>
> Thanks for checking out our work. It is worth noting that all experts are initialized with pre-trained weights. Therefore, the scenario of selecting a 'randomly initialized expert' does not arise :)

---

### Decision · Program_Chairs · 2025-09-17

**Decision:**

Accept (poster)

**Comment:**

This paper introduces a new VLA architecture - chatVLA-2, which aims to retain the open world reasoning capabilities of VLMs in VLA models, which are often lost during action prediction finetuning. The two major changes that chatVLA-2 makes are - 1) a switched a dynamic MoE backbone that helps alleviate catastrophic forgetting, 2) 2 stage training with co-training, followed by action finetuning lets action prediction be aligned with reasoning.  This methodology is then validated on two real world tasks - math matching and toy placement, which test whether the original VLMs reasoning capabilities are retained and combined with action prediction capabilities.

The reviewers generally appreciated the motivation and execution of the paper, but raised some valid concerns. Specifically, more explanation of *why* these design choices make such a difference, some additional baselines with simpler action-reasoning alignment recipes, validation on standard multimodal VLM prediction tasks, more formal description and validation of the claims (as suggested by reviewer chhG), some additional experiments on higher dexterity problems. These should be updated in the final vesion of the manuscript.

Overall the premise of the paper is valuable and it adds a useful datapoint to the literature on VLA models.